# Scenario Simulation of the Relationship between Land-Use Changes and Ecosystem Carbon Storage: A Case Study in Dongting Lake Basin, China

**DOI:** 10.3390/ijerph20064835

**Published:** 2023-03-09

**Authors:** Wenqiang Zhou, Jinlong Wang, Yu Han, Ling Yang, Huafei Que, Rong Wang

**Affiliations:** 1College of Forestry, Central South University of Forestry and Technology, Changsha 410004, China; 2College of Business, Central South University of Forestry and Technology, Changsha 410004, China; 3College of Economics and Management, Southwest University, Chongqing 400715, China; 4Hunan Sports Vacational College, Changsha 410019, China; 5College of Fumiture and Art Design, Central South University of Forestry and Technology, Changsha 410004, China

**Keywords:** land-use change, carbon storage, PLUS model, gray prediction, Dongting Lake Basin

## Abstract

High-frequency land-use changes caused by rapid economic development have become a key factor in the imbalance of carbon sequestration within regions. How to balance economic development and ecological protection is a difficult issue for regional planning. Studying the relationship between future land-use changes and ecosystem carbon storage (CS) is of important significance for the optimization of regional land-use patterns. The research used the gray prediction model and coupled the patch-generating land-use simulation (PLUS) model and the integrated valuation of ecosystem services and trade-offs (InVEST) model. On this basis, the evolution characteristics and spatial coordination between land-use changes and CS in the Dongting Lake Basin (DLB) in different scenarios in 2030 were simulated. The results show that: (1) The spatial distribution of CS remains stable in different scenarios, while land-use types with high carbon density in the periphery of cities are constantly invaded by construction land, which results in the greatest carbon loss in the urban areas. (2) Compared with the natural evolution scenario (NES), only 195.19 km^2^ of land-use types with high carbon density are transformed into construction land in the ecological protection scenario (EPS), generating a carbon sink gain of 182.47 × 10^4^ Mg. Conversely, in the economic development scenario (EDS), a total of over 1400 km^2^ of farmland and ecological land are transformed into construction land, which weakens the carbon sequestration capacity of ecosystems, and more than 147 × 10^4^ Mg of carbon loss occurs in the urban areas. (3) The planned development scenario (PDS) takes ecological protection and economic development both into consideration, which not only generates a carbon sink gain of 121.33 × 10^4^ Mg but also reduces the carbon loss in urban areas by more than 50%. The PDS performs well in both land use and CS growth and can better motivate the effect of land-use changes in increasing the carbon sink, which is also proved by analysis of the coordination between land-use intensity (LUI) and CS. Therefore, the PDS better satisfies the future development demand of DLB and can provide a reference for sustainable land use in the basin.

## 1. Introduction

Land-use changes, one of the most prominent features of economic development, have caused more than 60% of global carbon emissions since 1850 [1], and high-intensity land use has seriously affected the terrestrial carbon cycle [2]. In the National Land Planning Outline (2016–2030) released by China, driven by the excessive consumption of the economy and society, the intensification of human–land conflict has led to a significant degradation in the carbon storage (CS) function in local regions. Currently, the CS of terrestrial ecosystems in China is declining at a scale of 9.3 Tg per year due to land-use changes [3]. However, China is in a critical period of modernization development, and the demand for production and living land is expanding, and the blind development and disorderly expansion of cities will further accelerate carbon loss [4]. With the goal of “carbon peaking and carbon neutrality” in China, new requirements have been put forward for territorial planning programs, which also aim to stimulate the carbon sequestration and sink potential of terrestrial ecosystems. Therefore, scientific assessment of CS in terrestrial ecosystems based on territorial spatial master planning has become an important issue and is of great significance for carbon reduction.

There are significant differences in the carbon sequestration capacity of vegetation and soil in different land-use types [5], and the interconversion between land-use types will definitely cause an increase or decrease in CS [6]. However, the carbon sink effect of each land-use type conversion varies in different regions and even shows carbon loss. Since this carbon sink process is complex and variable, it exhibits different evolutionary characteristics in different spatial and temporal contexts [7,8]. Therefore, it is also important to clarify the dynamic relationship between land-use changes and CS.

CS in terrestrial ecosystems is mainly evaluated using field measurements, biomass inventories, and models. Therein, traditional methods such as field measurements and biomass inventories are difficult and costly to implement in large-scale and long-time-series studies [9], so researchers mainly use the model method based on land use. The integrated valuation of ecosystem services and trade-offs (InVEST) model has been widely used in the evaluation of CS due to its high operating efficiency, ease of access to data, and high accuracy [10,11]. On this basis, researchers have studied land-use changes and CS at different regional scales, including basins [5,11], administrative districts [12,13], and urban agglomerations [4,14], by combining scenario simulation and InVEST. Numerous studies have shown that higher and more stable CS can be realized by inhibiting blind urban expansion and relying on low land-use transition [13,15,16]. For example, the incremental carbon sequestration of forest ecosystems in Portugal would increase from 5.7% to 29.5% if the development limit of forest land were higher [13]. In Hawaii, USA, CS would increase by 0.5% by optimizing agricultural land and increasing the size of urban green space [15]. In its study of the Jiroft Plain in Iran, moreover, by comparing multiple planning scenarios, only environmentally friendly land planning policies would reverse the declining trend of CS and result in an average annual increase of 60.6 × 10^4^ Mg [16]. However, existing studies tend to focus more on the carbon sink effects of individual ecosystems or ecological protection policies and lack consideration of the complete terrestrial ecosystem and sustainable policies [17]. Furthermore, land-use changes are a multifactorial process, and the main challenge in balancing the terrestrial carbon cycle is to identify the best match between the many land planning options.

Although researchers have made much progress in the evaluation of regional CS, certain deficiencies remain. On the one hand, most researchers overlooked the obvious interannual volatility of carbon density and used fixed carbon density to estimate regional CS in different periods [18]. This attributes CS changes completely to land-use changes, and therefore, the evaluation results are highly uncertain. On the other hand, scenario models such as cellular automata (CA)–Markov, future land-use simulation (FLUS) model, and conversion of land use and its effects at small regional extent (CLUE-S) model run inefficiently and perform poorly in simulations when processing large-scale, high-resolution land-use data [19]. In particular, these models are likely to neglect dynamic evolution characteristics of ecological land (forests and grassland, for instance) [20,21]. To solve this problem, the present research makes the following contributions: (1) the patch-generating land-use simulation (PLUS) model developed using the High-Performance Spatial Computational Intelligence Lab of China University of Geosciences (HPSCIL@CUG) is introduced, which can better reflect the transition trend among various land-use types at the patch scale and therefore achieve high-accuracy simulation [21,22]; and (2) the gray prediction model is used to obtain carbon densities of the study area in multiple time periods. In addition, the PLUS and InVEST models are coupled to more accurately reflect the spatiotemporal and scenario evolution of ecosystem CS in the study area.

The Dongting Lake Basin (DLB), as one of the most important ecological nodes, has an important position in the Yangtze River Basin and even in China for its carbon sequestration and sink function [23]. However, due to the rapid expansion of large urban clusters in the basin, the high intensity and high frequency of land-use conversion has resulted in large areas of carbon loss and a serious imbalance between the economy and ecology [14,24]. Thus, the Main Functional Area Planning sets out the objectives of the rational allocation of land resources and the promotion of the green transformation of society in DLB.

In view of this, this study uses the PLUS + InVEST + GM(1,1) model to simulate and explore the impact of land-use changes on CS in DLB under territorial spatial master planning and its pathways of action. At the same time, the following three research objectives were set: (1) to explore the land-use patterns and their transformation characteristics in DLB under different scenarios, (2) to dynamically assess the dynamic changes in CS in the terrestrial ecosystem of DLB under different scenarios, and (3) to examine the response of land-use changes and CS in DLB under different scenarios and their matching coordination.

## 2. Materials and Methods

### 2.1. Study Region

The DLB (24°38′–30°24′ N, 107°16′–114°15′ E) is situated in the middle reaches of the Yangtze River and covers four sub-basins (Xiangjiang, Zijiang, Yuanjiang, and Lishui River Basins) as well as Dongting Lake area (Figure 1). The basin has an area of 2.63 × 10^5^ km^2^, which is about 14.6% of the total area of the Yangtze River Basin [25]. DLB has a humid subtropical monsoon climate, with an average annual temperature of 17 °C and an average annual precipitation of 1437 mm. Moreover, DLB is surrounded by mountains to the east, south, and west, hills in the center, and plains in the north, with the topography sloping from southwest to northeast, which breeds a huge and complete terrestrial ecosystem. However, during 1980–2020, the ecological risks in DLB became more and more serious, and the ecosystem service capacity has significantly decreased [23,24,25,26]. DLB is the core of economic development in central China. As of 2020, the total population of DLB was 73.3 million, with a per capita GDP of about 7 × 10^4^ yuan and an urbanization rate (57.2%) slightly lower than the national average (57.4%). Due to geographical constraints, the development level of the Xiangjiang River Basin and Dongting Lake area is significantly higher than that of other sub-basins. Meanwhile, in the past 40 years, a total of 2169.45 km^2^ of farmland and 1589.13 km^2^ of ecological land (forests, grassland, and wetland, etc.) in DLB were converted to new construction land to meet the land needs for economic development; however, only 73.68 km^2^ of ecological land was added. Moreover, uncontrolled land development and urban expansion are further enhancing.

### 2.2. Data Sources and Processing

The land-use data in the study area were derived from the Resource and Environmental Science Data Center of the Chinese Academy of Sciences (http://resdc.cn/ (accessed on 12 September 2022)) at a spatial resolution of 30 m. The data were divided into seven types: farmland, forests, grassland, wetland, waters, construction land, and unused land; because land-use changes are a result of the joint action of natural and artificial factors [18], the selected drivers included climate (annual mean temperature, annual precipitation, and annual hours of sunshine), environment (elevation, slope, soil type, and normalized difference vegetation index (NDVI)), distance (distances from road networks and waters), and economic society (per capita GDP and population density). Among them, the climatic factors were obtained by Kriging interpolation of data collected from the China Meteorological Data Service Center (http://data.cma.cn/wa (accessed on 12 September 2022)), and the spatial resolution output parameter was set to 30 m. The digital elevation model (DEM) data were derived from NASADEM (https://www.earthdata.nasa.gov/ (accessed on 12 September 2022)) at a spatial resolution of 30 m. Slope data were obtained by extraction and processing of DEM data. Data pertaining to soil type and economic society factors were both collected from the Resource and Environmental Science Data Center of the Chinese Academy of Sciences (http://resdc.cn/ (accessed on 12 September 2022)) at a spatial resolution of 1 km, and the soil type and economic society factors were resampled to 30 m by nearest neighbor method and bilinear method, respectively. NDVI data were provided by the GEE platform (https://earthengine.google.com/ (accessed on 12 September 2022)) and the Qinghai–Tibet Plateau National Scientific Data Center (https://data.tpdc.ac.cn (accessed on 12 September 2022)) at a spatial resolution of 30 m. Data pertaining to distance factors were obtained by Euclidean distance matrix analysis of Open StreetMap (https://www.openstreetmap.org/ (accessed on 12 September 2022)), and the spatial resolution output parameter was set to 30 m. In addition, the coordinate projection (Krasovsky_1940_Albers) and the number of rows and columns (22,584, 21,505) of all driving factors were unified with land-use data.

### 2.3. Methods

As shown in Figure 2, this study mainly included the following parts: (1) simulation and analysis of land-use changes, (2) assessment and analysis of dynamic changes in CS, and (3) response mechanism of land-use changes to CS. The method and analysis steps are described in detail below.

#### 2.3.1. Simulation of Land-Use Changes Based on the PLUS Model

In the Emissions Gap Report 2022 released by the United Nations Environment Programme, carbon emissions remain high, and climate disasters can only be avoided if system-wide transformation is urgently implemented before 2030. In addition, according to the Carbon Peaking Action Programme before 2030, the low-carbon transformation of urban and rural construction in China is to be promoted on the basis of optimizing the spatial development pattern of the country and the efficient use of land resources. Therefore, it is of great significance to explore land-use changes and its trends in 2030 to achieve a carbon peak.

The PLUS model is composed of the land expansion analysis strategy (LEAS) and CA model of multitype random patch seed (CARS) modules. Therein, the LEAS module obtains development probabilities of various land-use types by using the random forest algorithm; the CARS module simulates a land-use change pattern at the patch scale combining mechanisms of the traditional CA model under constraints of development probabilities of various land-use types [21]. The simulation of land-use changes of DLB in 2030 includes the following two aspects:(1)Verification of simulation accuracy

Development probabilities of various land-use types from 1980 to 2000 and 2000 to 2020 were obtained using the LEAS module based on land-use data of DLB in three periods (1980, 2000, and 2020). Then, taking land-use data in 1980 and 2000 as the base maps, the CARS module was employed to simulate land-use situations in 2000 and 2020. By verifying the accuracy through comparison with actual land-use data, it was calculated that the Kappa coefficients are 0.8915 and 0.9064, overall accuracies are 91.4% and 94.9%, and FOM values are 0.1043 and 0.1066, respectively. The result indicates that the selected drivers enable a favorable simulation effect and high accuracy, which meet the demand for simulating future land-use changes. Therefore, the land-use data in 2020 were used as the base map to simulate land-use changes in 2030 by selecting the optimal parameter set.

(2)Scenario setting of future land-use changes

Combining research experience of existing scenario simulations [9,21,27] with the ability of the PLUS model to achieve an optimal land-use structure under multiobjective programming [21], the natural evolution scenario (NES), ecological protection scenario (EPS), economic development scenario (EDS), and planned development scenario (PDS) were set. The simulation parameters and the transition matrix were determined by referring to the land-use transition probability from 1980 to 2020 and simulation accuracy.

NES: The scenario is based on the land-use data in three periods from 1980 to 2020 to predict the demand for different land-use types using the linear regression method and Markov chain. The scenario continues the historical land-use change trend in the study area, with no function-restricted areas and planned development areas.EPS: The scenario aims to reflect that in order to achieve the ecosystem restoration objective, the government in the study area intensifies enforcement intensity for ecological protection policies, stringently controls increases in construction land, and encourages returning farmland to forests, grassland, and lakes, as well as vegetation and wetland restoration. Therein, the function-restricted areas include ecological barriers including the Luoxiao–Mufu Mountains, Nanling Mountains, and Wuling–Xuefeng Mountains designated in the Territorial Spatial Master Planning of Hunan Province (2021–2035). Based on the NES, there are the following settings, apart from the function-restricted areas: (1) stringently restricting the transition of forests, grassland, wetland, and waters to other land-use types; (2) improving transition probabilities of farmland and unused land to forests, wetland, and waters by 60%, reducing transition probabilities of farmland and unused land to construction land by 80%, and improving the probability of transition of grassland to forests by 60%; and (3) setting a buffer of 10 km in the periphery of existing urban areas to meet the minimum demand for urbanization.EDS: The scenario gives priority to meeting production and living needs for socioeconomic development so that the demand for farmland and construction land grows substantially. The scenario mainly includes the following contents: (1) based on the NES, improving transition probabilities of all land-use types (except for waters) to farmland and construction land by 50%; (2) setting the southeast of Guizhou Province, the Yichang–Jingzhou–Jingmen–Enshi urban agglomeration, and the circum–Changsha–Zhuzhou–Xiangtan urban agglomeration designated in the Territorial Spatial Master Planning (2021–2035) and Main Functional Area Planning as planned development areas; and (3) setting waters in the basin in 2020 as function-restricted areas in an attempt to meet the demand for water for production and domestic use.PDS: The three aforementioned scenarios should coexist in a practical planning framework, necessitating trade-offs when planning the ecological, production, and living spaces. On the basis of development areas of urban agglomerations set in the EDS, the PDS also involves the following aspects: (1) setting the 1-hour commuting circle in the Development Plan for Changsha–Zhuzhou–Xiangtan Metropolitan Area as the buffer at the urbanization boundary; (2) setting ecological barriers such as the Wuling Mountains, Nanling Mountains, and Dongting Lake wetland designated in the Main Functional Area Planning and Comprehensive Water Environment Control Plan of Dongting Lake as function-restricted areas, in which transition from forests, wetland, and waters to other land-use types is prohibited; and (3) setting existing cultivated land in main agricultural producing areas of the basin in 2020 as function-restricted areas, to achieve the objective of protecting cultivated land and basic farmland. In addition, the development intensity of various land-use types in nonfunction-restricted areas is improved to 6.9% based on the NES according to the Main Functional Area Planning.

#### 2.3.2. CS Assessment of Terrestrial Ecosystems Based on the InVEST Model

(1)Estimate of CS

Using maps of land-use types and carbon density, the carbon storage and sequestration module of the InVEST model estimates the net amount of carbon stored in a land parcel over time [16,28]:(1)$$CS_{x}=\sum_{i=1}^{7}(C_{i-above}+C_{i-below}+C_{i-soil}+C_{i-dead})\times S_{xi}$$
where *CS_x_* is the amount CS of pixel *x*; *C_i-above_*, *C_i-below_*, *C_i-soil_*, and *C_i-dead_* represent the carbon density of aboveground biomass, belowground biomass, soil, and dead organic matter on land use *i*, respectively, all in Mg·ha^−1^; *S_xi_* is the area of land use *i* in pixel *x*; *i* = 1,…,7, representing farmland, forests, grassland, wetland, waters, construction land, and unused land, respectively.

(2)Estimate of carbon density

A key challenge when assessing carbon sequestration is setting carbon density. In this study, we estimated carbon density from five aspects for DLB: (1) We estimated the carbon density of farmland and grassland by a regression model between crop biomass density and average NDVI, and grassland aboveground biomass and maximum NDVI, respectively, according to the findings of Piao et al. [29] and Fang et al. [30]. (2) We modified the forest biomass carbon density (including aboveground, underground, and dead organic matter) and soil carbon density, referring to the national forest resource inventory data and the continuous biomass conversion factor method and using the climate correction model [31,32,33]. (3) The wetland carbon density is estimated based on the multiobjective regional geochemical survey [34,35,36] and the Monitoring and Assessment Report on Ecological Status of Dongting Lake Wetlands (2015–2020). (4) As the construction land and unused land are mostly impermeable surfaces and bare rock, this study set the biomass carbon density of the above two types of land as 0, and their soil carbon density was estimated according to the study by Xi et al. [34,35,36]. (5) Since the water body does not involve vegetation or has very little biomass, its carbon pools are 0 [37]. Based on these aspects, the carbon density of different land-use types was estimated (Table 1).

(3)Prediction of carbon density in future years based on GM(1,1)

The gray prediction model GM(1,1) was introduced to predict the carbon density of DLB in 2030, which was used as the basis for subsequent research. As a traditional dynamic gray prediction model, GM(1,1) can improve model accuracy using the gray differential model in the case of incomplete and inaccurate system information, and therefore, it realizes fuzzy quantitative prediction of future variables [38]. The estimated carbon density of DLB from 1980 to 2020 was substituted in GM(1,1) to predict the carbon density in 2030 (Table 1).

#### 2.3.3. Coordinating the Model between Land-Use Changes and CS

Land-use change affects the function and efficiency of the land, which, in turn, causes changes in ecosystem services. The coordinating model [39], modified using the conventional elastic coefficient, was introduced to quantitatively study the coordination between land-use intensity (LUI) and CS in DLB and the changes therein (Table 2). For the convenience of analysis, 3 km × 3 km grids were used for the generation of statistics related to LUI, CS, and their coordination in DLB. The calculation of the degree of coordination followed this principle [40,41]:(2)$$O_{k}=\frac{\left|\left(ALUI_{k}+ACS_{k}\right)/\sqrt{2}\right|}{\sqrt{ALUI_{k}{}^{2}+ACS_{k}{}^{2}}}$$
where *O_k_* is the coordinating index of grid *k*, *O_k_* ∈ [0,1]; the larger the *O_k_* is, the better the coordination between *LUI* and *CS*. *ALUI_k_* and *ACS_k_* represent the annual growth rates of *LUI* and *CS* of grid *k*. Among them, *LUI* is used to characterize the degree of disturbance of land-use patterns by human activities. In this study, with reference to the study of Zhuang et al. [42], the intensity levels of different land-use types were classified as 1 for unused land; 2 for forests, grassland, wetland, and waters; 3 for farmland; and 4 for construction land. The calculation of *LUI* followed this principle [42]:(3)$$LUI_{k}=\sum_{i=1}^{7}I_{i}\times Area_{ki}$$
where *LUI_k_* is the *LUI* of grid *k*, *I_i_* is the intensity level of land use *I*, *Area_ki_* is the area of the land use *i* in grid *k*, and *i* has the same meaning as Equation (1).

#### 2.3.4. Standard Deviational Ellipse Analysis

Standard deviational ellipse (SDE) mainly uses the center, oblateness, and rotation angle of output to measure and study the temporal variation trends of the central tendency, dispersion, and directional trend of a certain factor. Some research has applied SDE to reveal the spatial distribution and directional trend of coordination between two factors [43]. In the research, the ellipses for coordination of LUI and CS in different scenarios were computed to reflect change characteristics of the spatial distribution, centripetal force, and directional trend.

## 3. Results

### 3.1. Land-Use Simulation in Different Scenarios

Table 3 and Figure 3 demonstrate that compared with 2020, areas of various land-use types of DLB change to different extents in different scenarios in 2030. The areas of farmland and unused land are projected to shrink, areas of wetland, waters, and construction land enlarge, while those of forests and grassland fluctuate.

By visualizing the transition among various land-use types with the transformed area larger than 1 km^2^ (Figure 4), the transition of farmland and grassland to construction land will be the main land circulation direction; the circum–Changsha–Zhuzhou–Xiangtan urban agglomeration and the southeast of Guizhou Province are predicted to be areas with the most intense land-use changes, and the urban development areas characterized by the presence of construction land will show different expansion trends (Figure 3).

According to Figure 4a–d: (1) In the NES (Figure 4a), the areas of farmland, forests, grassland, and unused land separately shrink by 917.25 km^2^, 84.62 km^2^, 368.83 km^2^, and 0.29 km^2^; the areas of wetland, waters, and construction land separately increase by 150.26 km^2^, 93.52 km^2^, and 1127.21 km^2^. Construction land will expand mainly by invading farmland, grassland, and forests, while a small amount of construction land will also be transformed into farmland. (2) In the EPS (Figure 4b), the area of farmland is predicted to decrease by as much as 972.28 km^2^, the area of construction land will only increase by 195.19 km^2^, while areas of forests, grassland, wetland, and waters will increase substantially under stringent ecological protection policies such as returning farmland to forests and lakes. Farmland becomes the only land-use type that is transformed into others, while ecological land, including wetland, grassland, waters, and forests, will receive the largest area of transition from farmland. In the meantime, the proportion of transferred farmland absorbed by construction land will plummet to 20.04%. (3) In the EDS (Figure 4c), because production and living needs are maximized such that the development of ecological land is not restricted, the decreased amplitude of the farmland area is projected to slow down; the area of construction land will enlarge as high as 1451.20 km^2^ by 22.05%; the areas of forests, grassland, and unused land will maximally reduce by 169.23 km^2^, 553.25 km^2^, and 5.49 km^2^; the area of waters is predicted to show no increase. Influenced by preferential policies for economic development, construction land invades a great deal of land, in which farmland, grassland, and forests separately account for 50.66%, 37.66%, and 11.38%. (4) In the PDS (Figure 4d), economic development, farmland protection, and ecological protection are considered at the same time in the basin, which enables a more reasonable land-use pattern to be developed. Therein, the area reduction in farmland will be 908.07 km^2^, which is only larger than that in the EDS; apart from the EPS, the forests area also shows a positive increase of 47.12 km^2^; the area of construction land grows by 1162.84 km^2^, approaching the level in the NES. The land circulation direction in the PDS is similar to that in the NES. The difference is that the expansion of construction land does not destroy existing forests but relies on the transition of farmland and grassland. In addition, the area of construction land transformed into farmland is predicted to decrease by 52.03%.

### 3.2. Dynamic Changes in CS in Different Scenarios

Simulations using the PLUS + InVEST model indicate that CS of DLB in NES, EPS, EDS, and PDS will separately be 3125.70 Tg, 3129.60 Tg, 3124.15 Tg, and 3126.62 Tg in 2030 (Table 4), all increasing to different extents compared with that in 2020. As for various land-use types, forests and farmland sequestrate more than 93% of carbon in DTB, while unused land contributes the least to carbon sequestration, as its CS amounts to less than 1%. Moreover, CS of these three land-use types remains quasi-stable in different scenarios. CS of grassland and wetland both increase substantially by 6.53 Tg and 1.76 Tg separately in the EPS. CS of grassland continues its increasing trend (to 2.44 Tg) in the PDS, while that of wetland declines by 1.20 Tg to a level lower than that in the NES. Construction land is found to have the largest increase in CS. Except for the EPS, in which the CS of construction land is lower than 30% (only 14.60%), CS always exceeds 30% in other scenarios, particularly in the EDS, where it reaches 35.87%.

By combining Figure 5a–e, the spatial distribution of CS is shown to remain stable in DLB. This is manifested as a horseshoe-shaped distribution pattern with CS decreasing from high-value areas of large mountains in the east, south, and west of the basin to the hilly area of central Hunan Province and Dongting Lake. The low-value areas are mainly distributed in the Dongting Lake area and various urban regions, which shows the important guidance of the land-use pattern on the spatial distribution of CS.

Combining with the interannual variation (Figure 5f–i), regions with carbon losses are also concentrated in the Dongting Lake area and various urban regions, as evinced by: (1) In the NES (Figure 5f), the carbon loss mainly occurs in the Dongting Lake area (0.48 Tg) and the southeast of Guizhou Province (0.01 Tg), and CS in the Dongjiang Lake area also decreases by 0.002 Tg. (2) In the EPS (Figure 5g), ecosystems across the whole basin are effectively protected. Land-use types change in large areas in the Dongting Lake area due to the implementation of ecological restoration measures, such as returning farmland to forests and lakes and wetland restoration. This results in decreased carbon sequestration capacity in local regions and, therefore, a carbon loss of 0.51 Tg. (3) In the EDS (Figure 5h), production and societal activities are strongly supported across the whole basin so that the carbon loss risk rises substantially in regions with intense human activities. The Dongting Lake area, Changsha–Zhuzhou–Xiangtan metropolitan area, and the southeast of Guizhou Province are projected to have large carbon losses of 0.61 Tg, 0.55 Tg, and 0.07 Tg, respectively. Influenced by the development of the circum–Changsha–Zhuzhou–Xiangtan urban agglomeration, carbon losses exceeding 0.04 Tg will also occur in Hengyang, Changde, Loudi, Huaihua, and Zhuzhou in Hunan Province, as well as Anyuan in Jiangxi Province. (4) In the PDS (Figure 5i), the land-use pattern tends to be reasonable, and the carbon loss risk declines across the whole basin. The carbon losses of the Changsha–Zhuzhou–Xiangtan urban agglomeration and the southeast of Guizhou Province will separately be 0.17 Tg and 0.03 Tg, lower than 50% of those in the EDS. Because the Dongting Lake area possesses favorable locational conditions for production and living, human activity intensity always remains high, so the carbon loss in the area will also be high, reaching 0.59 Tg. Moreover, in places such as Hengyang in Hunan Province, which experience a carbon loss in the EDS, the amount of carbon loss will decrease, or the CS will increase from decreasing in the PDS.

### 3.3. The Relationship between Land-Use Changes and CS

#### 3.3.1. Influences of Land-Use Changes on CS

Different land-use types differ significantly in their carbon sequestration capacity, so land-use changes influence regional CS. To estimate influences of land-use changes on CS of DLB in different scenarios in 2030, CS changes were plotted (Figure 6).

Combining this with Figure 4: (1) In the NES (Figure 4a, Figure 6a), the transition between different land-use types will result in a carbon loss of 170.62 × 10^4^ Mg. To be specific, farmland, forests, and grassland are transformed into waterbodies and construction land so that the carbon sequestration capacity of vegetation and soil of the original land-use types is weakened. Although the wetland restoration from some farmland brings increases in CS by about 121.10 × 10^4^ Mg, it fails to complement the carbon loss induced by land-use changes. (2) In the EPS (Figure 4b, Figure 6b), the transition between different land-use types increases CS by 182.47 × 10^4^ Mg, mainly because the basin is oriented to ecological protection. This not only places stringent restraints on the transition of farmland, forests, and grassland to construction land but also drives the transition from land-use types with a low carbon density to those with a high carbon density, which stimulates the carbon sequestration potential. For example, farmland is transformed into forests, grassland, and wetland, which enables increases in CS by 106.71 × 10^4^ Mg, 15.52 × 10^4^ Mg, and 199.09 × 10^4^ Mg, respectively. (3) In the EDS (Figure 4c, Figure 6c), the transition trend between land-use types continues from the level in the NES, while the carbon loss reaches a higher level of 342.69 × 10^4^ Mg. This is mainly because the transition from forests and grassland with high carbon sequestration capacity to construction land enlarges by more than 50%. The second cause is that transition from farmland to wetland reduces substantially, so the resulting increment in CS decreases correspondingly. (4) In the PDS (Figure 4d, Figure 6d), only 92.08 × 10^4^ Mg of CS is lost, which is much lower than the CS losses in the NES and EDS. The transition from farmland and forests to construction land results in a CS loss of 182.10 × 10^4^ Mg. Despite this, the increment in CS attributed to the transition from farmland and grassland to forests and wetland with higher carbon sequestration capacity and clearing out some construction land is as much as 90.53 × 10^4^ Mg. This compensates for the CS loss caused by the transition between different land-use types.

#### 3.3.2. Spatial Coordination between Land-Use Changes and CS in Different Scenarios

As shown in Figure 7, the land-use changes are adapted (CS growth ahead of improvement in LUI) with CS changes in DLB in different scenarios. That is, increases in LUI and CS are in a stage of adaptation, in which the carbon sequestration capacity of ecosystems in the basin manages to adapt to rapid socioeconomic development; however, the carbon loss induced by intensive land use cannot be compensated in case of any carelessness in development (or accidents). In regions such as the Dongting Lake area, the circum–Changsha–Zhuzhou–Xiangtan urban agglomeration, and the southeast of Guizhou Province, the coordination between LUI and CS shows significant spatial heterogeneity (Figure 7). This is manifested as follows: (1) In the NES (Figure 7a), the lack of coordination (CS growth ahead of improvement in LUI) is mainly distributed in wetland and forests with low intensity of human activities in the Dongting Lake area. In these regions, the stability of these ecosystems is ensured, and the carbon sequestration capacity is improved under few artificial disturbances. LUI and CS are coordinated in most urban agglomerations, which means that LUI and CS can grow synergistically. (2) In the EPS (Figure 7b), the carbon sequestration capacity of ecosystems is enhanced substantially, while land-use and development intensity are strictly controlled. This leads to a surge in uncoordinated regions (CS growth ahead of improvement of LUI), which are concentrated in the Dongting Lake area, the southeast of Guizhou Province, and the banks of Yuanjiang River. In comparison, coordinated regions appear sporadically in the Xiangjiang River Basin. (3) Contrary to the situation in the EPS, a large area of ecological land is transformed into construction land in the EDS (Figure 7c). This reduces the carbon sequestration capacity of the whole basin and gives rise to circular uncoordinated regions (CS growth lagging behind improvement in LUI) appearing in the periphery of cities with high LUI. Other regions with concentrated artificial disturbances are also found to be adapted or coordinated, both with CS growth lagging any improvement in LUI. (4) In the PDS (Figure 7d), the lack of coordination between increments of LUI and CS is rare, and such a phenomenon is mainly concentrated in the Wuling district in Changde City, Hunan Province (uncoordinated with CS growth lagging any improvement in LUI). The periphery of other cities is coordinated: this indicates that the ecosystems and economic society develop in good coordination in DLB in the scenario, especially urban agglomerations, while the ecological risk caused by excessively rapid economic development also cannot be overlooked.

Further analysis shows that ellipses in the SDE analysis are distributed along the northeast to southwest direction in different scenarios; however, the four ellipses differ significantly in their spatial locations and coverages (Figure 7). The minor semiaxes of ellipses for the coordination between LUI and CS in the NES, EPS, EDS, and PDS are 28.06 km, 41.95 km, 18.48 km, and 40.58 km; the oblateness is 56.12 km, 83.90 km, 36.97 km, and 81.17 km; and coverages are 148,736.14 km^2^, 128,876.02 km^2^, 150,527.41 km^2^, and 146,430.71 km^2^, respectively. Compared with the NES, the ellipses in the other scenarios show the following characteristics: (1) The coverage is the largest in the EDS, while the center shifts to the south. In addition, due to the long minor semiaxis and small oblateness of the ellipse, the coordination has a higher dispersion and insignificant directional trend of spatial distribution. (2) The ellipses in the EPS and PDS have similar minor semiaxes and oblateness. Although the aggregation of the degrees of coordination in the two scenarios is inferior to that in the NES, the directional trends in the two scenarios are more explicit. Combining with the coverage, the ellipse has the minimum area in the EPS, and its center lies in the upper and middle reaches of the basin. The ellipse in the PDS has a smaller area than that in the NES, while its center is located on the line between the southeast of Guizhou Province and the circum–Changsha–Zhuzhou–Xiangtan urban agglomeration. This indicates that the land-use modes in regions with concentrated human activities of the basin play a key role in the carbon sequestration of ecosystems.

## 4. Discussion

### 4.1. The Relationship between Land-Use Changes and CS

The transition between land-use types with low and high carbon densities has both positive and adverse effects on the carbon sequestration of ecosystems [44]. On the one hand, enlarging the areas of farmland and construction land will lead to carbon losses. On the other hand, ecological protection and restoration measures such as returning farmland to forests or grassland, as well as reclamation and vegetation restoration of construction land, will increase the carbon sink, making it necessary to minimize the carbon loss caused by land-use changes while meeting socioeconomic development demands.

Comparing the results of this study with existing studies on the Yangtze River Basin [24] and Poyang Lake Basin [45], the future land-use trend in DLB is represented by a gradual transition from farmland and ecological land (mainly forests and grassland) to new construction land, and the transition will be concentrated in existing urban areas, which is certain. The difference, however, is that farmland in DLB will also be shifted mainly to wetland and waters, with a certain scale of fallowing only occurring during EPS and PDS. This may be due to the concentration of farmland in the Dongting Lake area and along the shores of the tributaries, where returning farmland to lakes and wetland restoration is a more appropriate ecological protection measure. Secondly, the negative effect of human intervention in the DLB is significant, with large spatial differences in carbon sequestration capacity. This is mainly due to the large urban clusters that have been formed during the economic construction of “emphasizing development and neglecting protection” [14] and which are scattered in sub-basins.

The research also reveals that the invasion of construction land to land-use types with high carbon density is the key to carbon loss. Although the transition from farmland, construction land, and unused land to forests, grassland, and wetland can greatly improve CS, land-use changes in DLB are dominated by the transition from farmland and grassland to construction land. This is unfavorable for increasing CS in the basin. The PDS based on the Territorial Spatial Master Planning (2021–2035) and Main Functional Area Planning is significantly better than other scenarios. Under the PDS, the area of forests will increase, and the expansion of construction land will no longer destroy existing forested land, relying on the transfer of farmland and grassland. Moreover, the risk of carbon loss from major urban agglomerations will be significantly reduced, and the carbon sequestration capacity of the DLB will be increased, which is similar to the results of the SSP1–2.6 scenario set by Zhang et al. [24] in the Yangtze River Basin. The comparative analysis of changes in coordination between LUI and CS further proves that due to reasonable land-use changes, the PDS can meet CS growth and social development. It also stimulates the transition from farmland and grassland to forests and wetland, thus enhancing the ability of soil to sequester carbon.

### 4.2. Policy Implications and Optimization Suggestions

How to realize the sustainable development of ecology and economy is the core problem to be solved urgently in territorial spatial master planning. The Carbon Peaking Action Programme before 2030 emphasizes the protection of ecosystems and the strict prohibition of large-scale development. However, in some parts of China, land planning schemes consistently serve economic development [46], and these schemes fail to balance ecological protection with economic development. We believe that, in the future, if ecological–economic sustainability is to be achieved, forest destruction and wetland degradation caused by human activities must be prevented. In forest areas, the policy of returning farmland to forests should be implemented with greater vigor, not only to effectively improve the carbon sequestration capacity of forest ecosystems but also to maintain biodiversity and regulate climate, etc. Moreover, the policy of returning farmland to lakes and restoring wetland and vegetation needs to be seriously implemented, as good water ecosystems are vital to human society and terrestrial ecosystems [25]. In addition, rational optimization of the internal spatial pattern of built-up areas can effectively improve the efficiency of land resource use and restrain urban sprawl [45].

In summary, the following suggestions for optimizing the future land-use pattern in DLB are recommended: (1) The scale of the expansion of urban agglomerations within the Xiangjiang River Basin and the Dongting Lake area should be strictly controlled, and in particular, the land-use pattern of the Changsha–Zhuzhou–Xiangtan metropolitan area should be optimized. In regions such as western Hunan, which is relatively economically backward, the scale of the expansion of construction land can be appropriately controlled, but the development intensity of other land-use types should be reduced. (2) It is suggested to set conservation areas in forests and wetland within urban agglomerations, such as Yuelu Mountain, Hengshan Mountain, and Dongting Lake wetlands, to enhance the carbon sequestration capacity of urban ecosystems and improve the quality of the human habitat. (3) It needs to enhance the protection of high-quality farmland in the Dongting Lake area, actively promote the transition from poor and barren land around urban and mountainous areas to ecological land, and develop unused land suitable for culturing forests and grassland or reclamation.

### 4.3. Accuracy of Estimation Results of CS

The research is different from most existing studies that directly use data summarized from the literature, in which carbon density is determined based on remote sensing inversion and model correction. Considering the interannual variation in carbon density, the gray prediction model was adopted to attain carbon density values in future years, which, to some extent, guarantees the accuracy of the estimation results of CS. The measured or simulated carbon densities of various ecosystems in DLB were collected and then compared with the current research results (Table 5). The carbon densities of aboveground biomass and underground biomass of forest vegetation in DLB collected from the dataset on carbon density in Chinese terrestrial ecosystems in 2010 [47] are approximate to those obtained in the present research. The carbon density of forest soil shows a certain deviation from that attained in the present research. The carbon density of arable soil is consistent with that acquired in the present research. The carbon density of grassland is approximate to that in the present research. Carbon densities of impervious surfaces and other land-use types conform to those of the construction land and unused land obtained herein. Moreover, the average carbon density of Dongting Lake wetland biomass and the carbon density of soil measured by Zhang et al. [48] and Kang et al. [49] are both akin to the carbon densities of wetland biomass and soil obtained in the present research. Furthermore, the carbon density of wetland biomass is projected to decline, which agrees with the conclusion suggesting a decreased carbon sequestration capacity of Dongting Lake wetland mentioned in the Ecological Monitoring and Assessment Report of Dongting Lake Wetland (2015–2020). In summary, the carbon densities obtained in the research are reliable and can be used in the current research.

### 4.4. Limitations

(1) Although remote sensing inversion and model correction methods are used to determine carbon density values, which are efficient and scientifically sound for large regional scale studies, it may affect the accuracy of CS assessment compared to field experiments [29]. Even though dynamic carbon density has been used to evaluate CS in DLB, the limitation of the InVEST model that simplifies carbon sequestration still cannot be completely overcome. In particular, the model overlooks differences and dynamic changes in carbon density inside a single ecosystem. Future research should determine the dynamic change trend of carbon density in the study area by combining with field measurements and verify the trend by combining with Eddy flux to further improve the accuracy of CS evaluation results.

(2) The PLUS model is adopted to compensate for deficiencies in conventional models in determining drivers for land-use transition and in simulating the spatiotemporal evolution of landscapes [21]. The model also improves the simulation accuracy for changes in various land-use types. However, certain limitations remain: (1) The future land-use demands in different scenarios are all obtained by adjusting the NES according to relevant policy documents. Although differences in the development intensity and transition probability of various land-use types in different scenarios are considered, it fails to reflect the close relationship between future land-use changes and regional economic development. How to improve the rationality and accuracy of predictions of future land-use demands in different scenarios by combining them with the prevailing trends in economic development will become one of the key problems to be solved in future land-use simulations. (2) Regional development policies have strong guidance on land-use changes. Although the function-restricted areas and planned development areas are set according to relevant policies, the setting is not comprehensive enough, which, to some extent, influences the accuracy of the simulation. How to better achieve quantification and space expression of policy and institutional drivers is key to driver selection in future land-use simulations.

## 5. Conclusions

The PLUS model was used to simulate the land-use patterns of DLB in different scenarios in 2030. The PLUS model was also coupled with the InVEST model to evaluate the CS of ecosystems in different scenarios. The following conclusions are drawn:

(1) Compared with 2020, other land-use types exhibit similar trends in different scenarios in 2030, except for forests and grassland, which show different changes in different scenarios. Construction land in regions such as the Dongting Lake area, the circum–Changsha–Zhuzhou–Xiangtan urban agglomeration, and the southeast of Guizhou Province changes in a complex manner. In addition, future land-use changes will be dominated by the transition from farmland and grassland to construction land.

(2) CS across DLB will always increase in different scenarios in 2030. CS is distributed in a horseshoe-shaped pattern that decreases in a step-by-step manner from three directions (the east, south, and west) to the center and north. Regions with more rapid urbanization and more frequent human activities are found to have more significant dynamic changes in CS. Construction land invades large areas of land with high carbon sequestration capacity, which will induce carbon loss and weaken the ability of the whole basin in carbon sequestration. Moreover, land-use changes are a “double-edged sword.” The PDS can not only balance the advantages of EPS and EDS but also compensate for disorderly development in the NES to facilitate the development of the whole basin at the cost of a small carbon loss.

(3) LUI and CS growth are adapted (CS growth ahead of improvement in LUI) in DLB. Although the rate of growth of CS is higher than the improvement rate of LUI, the two are marginally coordinated. Compared with other scenarios, the coordination of the two has a more explicit directional trend in the PDS and a higher degree of coordination. Moreover, the PDS emphasizes the coordinating and promoting effect of reasonable land development on the carbon sequestration capacity of ecosystems in the basin.

## Figures and Tables

**Figure 1 ijerph-20-04835-f001:**
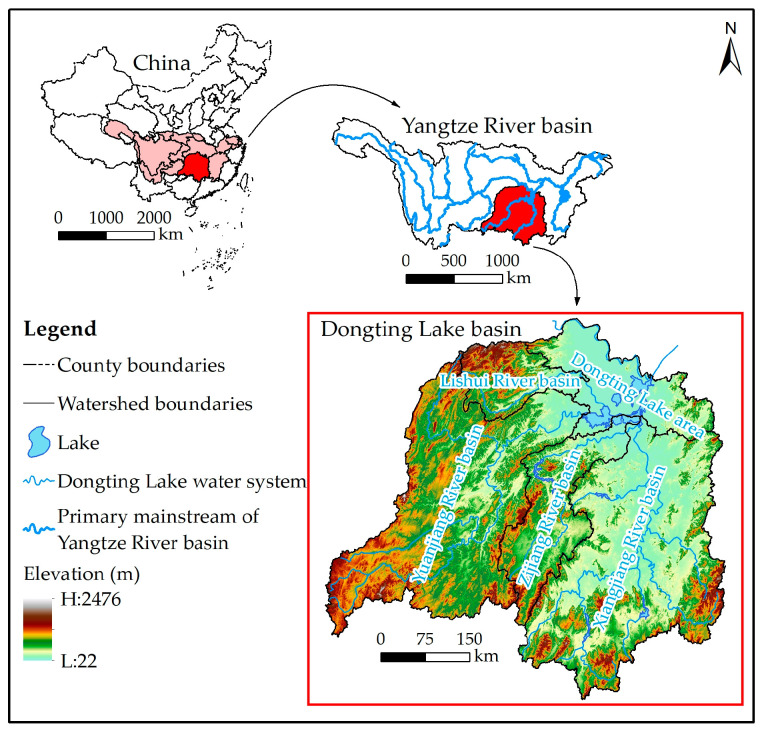
Location and terrain map of the Dongting Lake Basin (DLB, map of China from China Standard Map Service GS (2020) 4619).

**Figure 2 ijerph-20-04835-f002:**
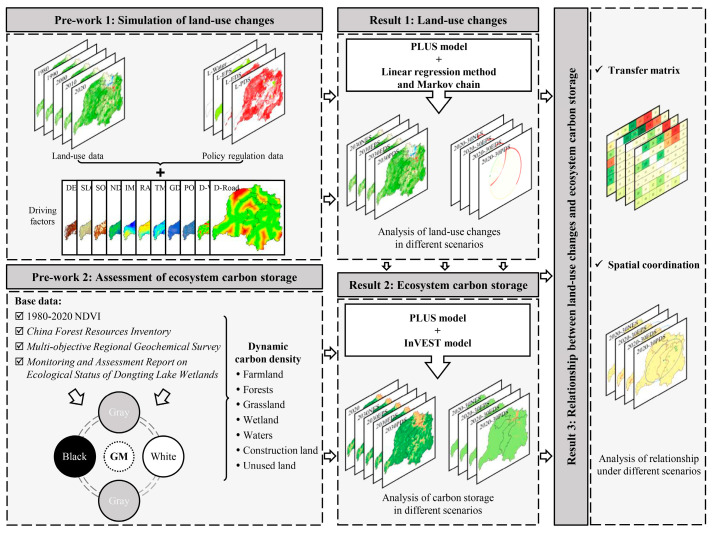
Flowchart of the methodology and analysis in this study.

**Figure 3 ijerph-20-04835-f003:**
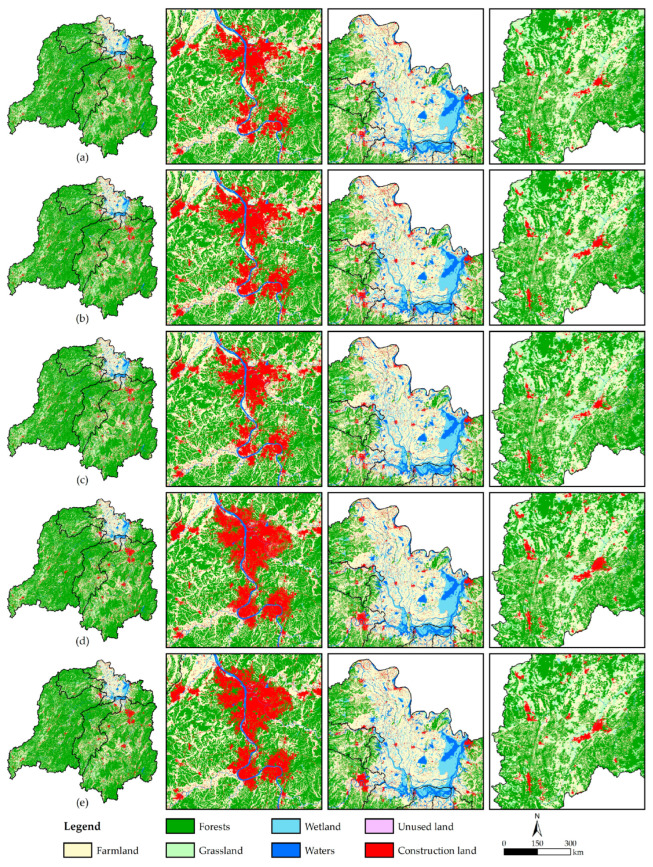
Spatial distribution of land-use types in DLB from 2020 to 2030. (**a**–**e**) In 2020/2030NES/2030EPS/2030EDS/2030PDS.

**Figure 4 ijerph-20-04835-f004:**
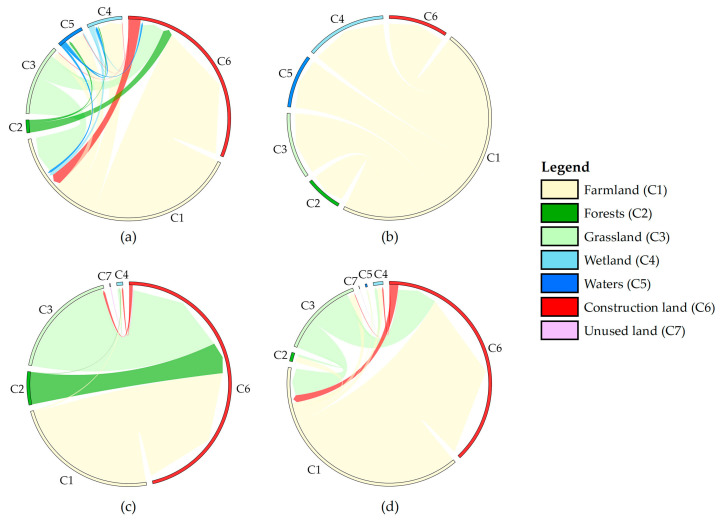
Transfer relationship of different land-use types in DLB from 2020 to 2030. (**a**–**d**) In 2020–2030NES/2020–2030EPS/2020–2030EDS/2020–2030PDS.

**Figure 5 ijerph-20-04835-f005:**
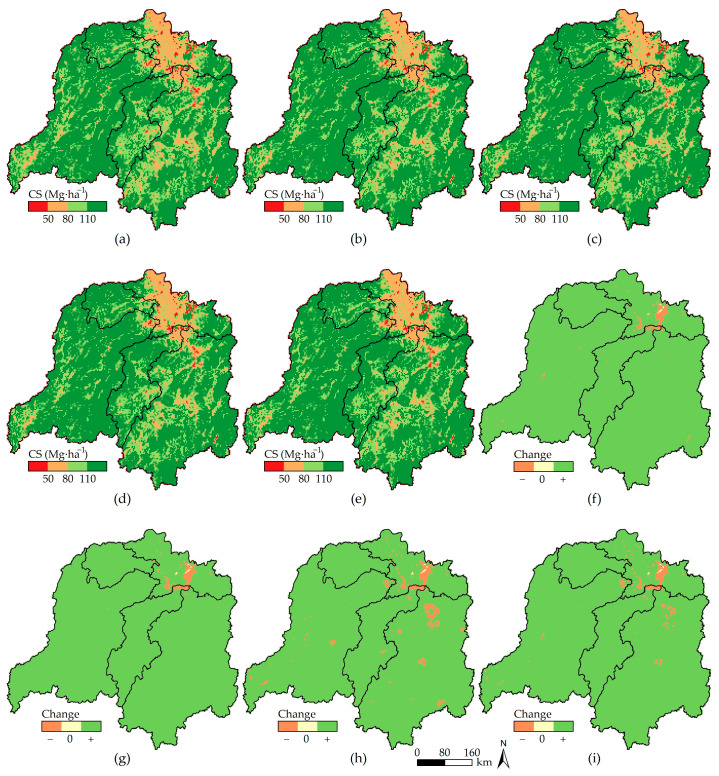
CS and its changes in DLB from 2020 to 2030. (**a**–**e**) Spatial distribution of CS in 2020/2030NES/2030EPS/2030EDS/2030PDS; (**f**–**i**) changes in CS in 2020–2030NES/2020–2030EPS/2020–2030EDS/2020–2030PDS.

**Figure 6 ijerph-20-04835-f006:**
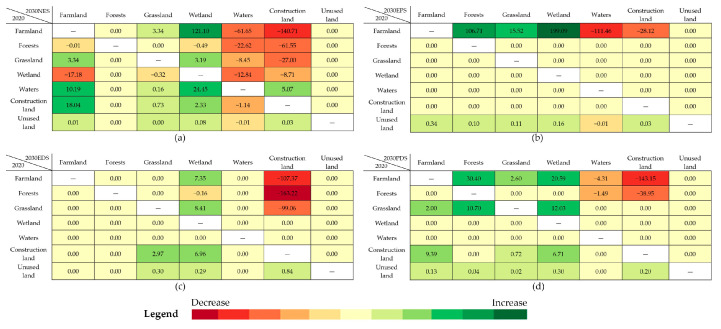
Transfer relationship of CS in DLB from 2020 to 2030 (unit: 10^4^ Mg). (**a**–**d**) In 2020–2030NES/2020–2030EPS/2020–2030EDS/2020–2030PDS.

**Figure 7 ijerph-20-04835-f007:**
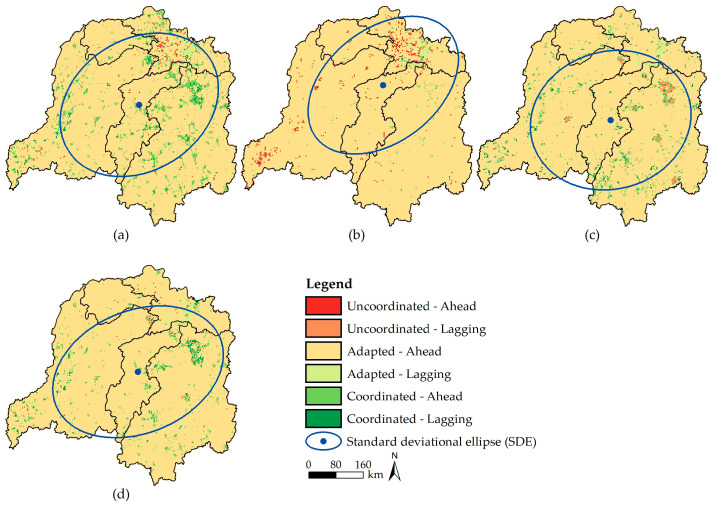
Coordination degree and standard elliptic difference between LUI and CS in DLB from 2020 to 2030. (**a**–**d**) In 2020–2030NES/2020–2030EPS/2020–2030EDS/2020–2030PDS.

**Table 1 ijerph-20-04835-t001:** Carbon density of different land-use types in DLB (unit: Mg·ha^−1^).

Land-Use Type	2020 Carbon Density				2030 Carbon Density			
	*C_i-above_*	*C_i-below_*	*C_i-soil_*	*C_i-dead_*	*C_i-above_*	*C_i-below_*	*C_i-soil_*	*C_i-dead_*
Farmland	1.80	0.35	61.41	0.00	1.93	0.38	66.06	0.00
Forests	27.12	7.32	111.12	1.16	28.05	7.57	114.96	1.20
Grassland	1.19	2.37	63.42	0.06	1.26	2.49	66.90	0.07
Wetland	8.50	1.95	131.61	0.95	8.24	1.89	127.62	0.92
Construction land	0.00	0.00	44.15	0.00	0.00	0.00	49.14	0.00
Unused land	0.00	0.00	30.47	0.00	0.00	0.00	29.94	0.00

**Table 2 ijerph-20-04835-t002:** Coordination types and relationships between land-use intensity (LUI) and carbon storage (CS).

*O*	Judgment Condition	Coordination Type		Relationships Between LUI and CS
[0,0.5)	ALUI < ACS	Uncoordinated	Ahead	LUI and CS are uncoordinated, and CS growth is ahead of improvement in LUI
	ALUI > ACS		Lagging	LUI and CS are uncoordinated, and CS growth lags improvement in LUI
[0.5,0.8)	ALUI < ACS	Adapted	Ahead	LUI and CS are managed so as to coordinate, and CS growth is ahead of improvement in LUI
	ALUI > ACS		Lagging	LUI and CS are managed so as to coordinate, and CS growth lags improvement in LUI
(0.8,1]	ALUI < ACS	Coordinated	Ahead	LUI and CS are coordinated, and CS growth is ahead of improvement in LUI
	ALUI > ACS		Lagging	LUI and CS are coordinated, and CS growth lags improvement in LUI

**Table 3 ijerph-20-04835-t003:** Area and change in land-use types in DLB from 2020 to 2030 (unit: km^2^).

	Year	Farmland	Forests	Grassland	Wetland	Waters	Construction Land	Unused Land
Area	2020	73,615.78	160,449.22	13,568.41	4416.96	4561.83	6580.18	30.46
	2030 NES	72,698.53	160,364.60	13,199.58	4567.22	4655.35	7707.39	30.17
	2030 EPS	72,643.50	160,570.26	13,785.44	4682.18	4737.22	6775.37	28.87
	2030 EDS	72,861.43	160,279.99	13,015.16	4448.08	4561.83	8031.38	24.97
	2030 PDS	72,707.71	160,496.34	13,207.69	4468.54	4570.83	7743.02	28.71
Change	2020–2030 NES	−917.25	−84.62	−368.83	150.26	93.52	1127.21	−0.29
	2020–2030 EPS	−972.28	121.04	217.03	265.22	175.39	195.19	−1.59
	2020–2030 EDS	−754.35	−169.23	−553.25	31.12	0.00	1451.20	−5.49
	2020–2030 PDS	−908.07	47.12	−360.72	51.58	9.00	1162.84	−1.75

**Table 4 ijerph-20-04835-t004:** CS in DLB from 2020 to 2030 (unit: Tg).

	Year	Farmland	Forests	Grassland	Wetland	Construction Land	Unused Land	Tot
Area	2020	467.90	2354.11	90.96	63.17	29.05	0.10	3005.29
	2030 NES	497.04	2434.01	93.35	63.33	37.87	0.10	3125.70
	2030 EPS	496.66	2437.14	97.49	64.93	33.29	0.09	3129.60
	2030 EDS	498.15	2432.73	92.04	61.68	39.47	0.08	3124.15
	2030 PDS	497.10	2436.01	93.40	61.97	38.05	0.09	3126.62
Change	2020–2030 NES	29.14	79.90	2.39	0.16	8.82	0.00	120.41
	2020–2030 EPS	28.76	83.03	6.53	1.76	4.24	−0.01	124.41
	2020–2030 EDS	30.25	78.62	1.08	−1.49	10.42	−0.02	118.86
	2020–2030 PDS	29.20	81.90	2.44	−1.20	9.00	−0.01	121.33

**Table 5 ijerph-20-04835-t005:** Comparison of carbon density of different land-use types (unit: Mg·ha^−1^).

Land-Use Type	Carbon Density	The Present Research	Reference Value	Data Source
Farmland	*C_i-soil_*	61.41~66.06	65.20	Xu et al. [47]
Forests	*C_i-above_*	27.12~28.05	29.58	Xu et al. [47]
	*C_i-below_*	7.32~7.57	10.40	Xu et al. [47]
	*C_i-soil_*	111.12~114.96	158.40	Xu et al. [47]
Grassland	*C_i-soil_*	63.42~66.90	68.20	Xu et al. [47]
Wetland	*C_i-above_ + C_i-below_ + C_i-dead_*	11.05~11.40	14.95	Zhang et al. [48]; Kang et al. [49]
	*C_i-soil_*	127.62~131.61	139.4~157.71	Zhang et al. [48]; Kang et al. [49]
Others	*C_i-soil_*	29.94~49.14	27.46~44.51	Xu et al. [47]

## Data Availability

Not applicable.

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
