# Peer review of "Scenario Simulation of the Relationship between Land-Use Changes and Ecosystem Carbon Storage: A Case Study in Dongting Lake Basin, China"

_ijerph, 2023, doi:10.3390/ijerph20064835_

Round 1

Reviewer 1 Report

This study explored the relationship between land use change and ecosystem carbon storage in Dongting Lake Badin under different scenario in 2030. It is significance of providing scientific information for sustainable development of Dongting Lake areas. However, some comments should be addressed.

1. In the abstract, I think the authors should use one or two sentences to introduce the study background. Using some digits to describe the results. Besides, the authors should check the expression of the sentences in line 17-20. Abbreviations and full spellings need to correspond, e.g. regional land use pattern simulation (PLUS).\

2. The abbreviations should be spelled out when first appeared, e.g. HPSCIL@CUG in line 72, please check this error throughout the paper.

3. The study aims and significance of this study should be introduced in the introduction part. Besides, the reason of choosing Dongting Lake should be introduced in the introduction part.

4. In the 2.1 part, the socio-economic, natural environment situation and land use change trends between 1980 and 2020 should be introduced.

5. The authors used different data with different resolutions in this study. The data processing information should be described in this part, e.g. how to unify the data resolution.

6. Reasons of choosing 2030 as the predicated year should be introduced in 2.3.1 part.

7. How to define the land use intensity.

8. Differences among the sub-figures in Fig 3 is not obvious. Using the land use change map to show it?

9. Policy implication according to the study results should be proposed in the discussion part.

10. I think it would better to define the abbreviation of carbon storage as CS, rather than Carbon.

Reviewer 2 Report

It's an pleasure to go through the research work and suggestions aimed at overall improvement of the manuscript. Authors have used InVEST model and landuse pattern simulation to depict a scenario of landuse changes on ecosystem carbon storage. I find the article novel and research question unique. Some comments,

1. Introduction is more focused on the models used rather than addressing the basic questions ' why we need such study?' ' how this work is unique?' 'how this work will impact the overall knowledge in this domain?' introduction is glimps of the overall what reader could expect in an article, hence to present the work international case studies can be brought up up in this section. In line number 77-83, the introduction should end with an concrete objective or a strong research question or a hypothesis.

2. Line number 89-90, in the sentance a phrase has been used "two kidneys", which may reffer to the importance of this ecosystem to the readers. But, it is not clear what this actually means and for an international reader this may have a variable meaning, which is not desired in this article. I would suggest to elaborate on this.

3. This article focused on remote sensing data to estimate the carbon,this may result in errors in carbon estimation in comparison to a field based on ecological survey. As the main focus of the study is on RS, obviously the authors would not be engaging in an field ecological survey. But this limitation need to be put into writting in the method part.

4. Rest of the article is excellently written and I will like to appreciate authors hardwork. These minor comments are made for overall improvement of the article.

Round 2

Reviewer 1 Report

The authors have addressed all the comments I proposed.